# Dynamics of Chromatin Opening across Larval Development in the Urochordate Ascidian *Ciona savignyi*

**DOI:** 10.3390/ijms25052793

**Published:** 2024-02-28

**Authors:** Muchun He, Yuting Li, Yajuan Li, Bo Dong, Haiyan Yu

**Affiliations:** 1Fang Zongxi Center for Marine EvoDevo, MoE Key Laboratory of Marine Genetics and Breeding, College of Marine Life Sciences, Ocean University of China, Qingdao 266003, China; hmc20160105@126.com (M.H.); liyuting6876@stu.ouc.edu.cn (Y.L.); lyj930722@126.com (Y.L.); bodong@ouc.edu.cn (B.D.); 2Liaoning Key Laboratory of Marine Animal Immunology, Dalian Ocean University, Dalian 116023, China; 3College of Animal Science, Inner Mongolia Agricultural University, Hohhot 010018, China; 4Laboratory for Marine Biology and Biotechnology, Qingdao Marine Science and Technology Center, Qingdao 266237, China; 5MoE Key Laboratory of Evolution & Marine Biodiversity, Institute of Evolution & Marine Biodiversity, Ocean University of China, Qingdao 266003, China

**Keywords:** ascidian, ATAC-seq, gene expression, open chromatin, RNA-seq, transcription factor

## Abstract

Ascidian larvae undergo tail elongation and notochord lumenogenesis, making them an ideal model for investigating tissue morphogenesis in embryogenesis. The cellular and mechanical mechanisms of these processes have been studied; however, the underlying molecular regulatory mechanism remains to be elucidated. In this study, assays for transposase-accessible chromatin using sequencing (ATAC-seq) and RNA sequencing (RNA-seq) were applied to investigate potential regulators of the development of ascidian *Ciona savignyi* larvae. Our results revealed 351 and 138 differentially accessible region genes through comparisons of ATAC-seq data between stages 21 and 24 and between stages 24 and 25, respectively. A joint analysis of RNA-seq and ATAC-seq data revealed a correlation between chromatin accessibility and gene transcription. We further verified the tissue expression patterns of 12 different genes. Among them, *Cs*-matrix metalloproteinase 24 (*MMP24*) and *Cs*-krüppel-like factor 5 (*KLF5*) were highly expressed in notochord cells. Functional assay results demonstrated that both genes are necessary for notochord lumen formation and expansion. Finally, we performed motif enrichment analysis of the differentially accessible regions in different tailbud stages and summarized the potential roles of these motif-bearing transcription factors in larval development. Overall, our study found a correlation between gene expression and chromatin accessibility and provided a vital resource for understanding the mechanisms of the development of ascidian embryos.

## 1. Introduction

The ascidian *Ciona savignyi* belongs to the urochordate subphylum [1,2,3], which comprises the closest relatives of vertebrates [4,5,6,7]. *Ciona* is an ideal model organism for studying tissue formation during chordate embryonic development [8]. Its early embryonic processes are relatively simple, its transgenic technology has been developed [9], and its genome has been sequenced [10,11] and deposited in the EDomics database [12]. Distinct morphological changes occur in various tissues of the ascidian from the early to the late tailbud stages, including tail bending [13] and epidermal and muscle cell extension, and notochord cells also undergo a series of complex morphological events, including elongation [14], migration, lumen formation [15,16,17,18], and lumen fusion [14,19,20]. The cytological process of ascidian tailbud development has been extensively studied, but the molecular mechanism of its regulation is poorly understood.

Assay for transposase-accessible chromatin with high-throughput sequencing (ATAC-seq) is a highly efficient technology for detecting accessible open chromatin [21]. In the last decade, 65 species have been subjected to ATAC-seq, and more than 2000 datasets have been generated, including data for protozoa, fungi, plants, and animals [22]. Moreover, ATAC-seq is useful for analyzing marine invertebrate embryos [23]. Ascidian chromosomal loci exhibit time-dependent dynamic changes in chromatin accessibility and conserved open chromatin regulatory elements around developmental regulatory genes [24], which are similar to those observed in *Drosophila* [25], *Caenorhabditis elegans* [26], and vertebrates [27]. However, the key regulatory factors in tailbud development remain unclear.

In our current work, we investigated accessible chromatin and the expression profiles of genes at three different developmental stages of *C. savignyi*, namely stages 21 (tail is 1-2-fold longer than the trunk and curves ventrally, complete with the intercalation of notochord cells), 24 (notochord vacuolation begins, palps start to be visible at the front end of the embryo, tail straightens), and 25 (ocellus melanization, all notochord cells have vacuoles), using ATAC-seq and RNA sequencing (RNA-seq). An integrated analysis of ATAC-seq and RNA-seq data was performed to identify changes in chromatin accessibility that influence gene expression as well as key regulators in the development of ascidian tailbuds, such as Krüppel-like factor 5 (*KLF5*) and matrix metalloproteinase 24 (*MMP24*), which regulate notochord lumen formation and expansion. We further identified nine transcription factors (t-box transcription factor *15/18/22* [*Tbx15/18/22*], forkhead box protein e [*FoxE*], forkhead box d-*b* [*FoxD-b*], six homeobox 3/6 [*Six3/6*], forkhead box i-a[*FoxI-a*], forkhead box protein k [*FoxK*], gata binding protein-a [*GATA-a*], fosb proto-oncogene [*Fos-B*], and forkhead box l2 [*FoxL2*]) that potentially affect ascidian development. These findings provide useful resources for further investigation of the mechanisms underlying ascidian larval development.

## 2. Results

### 2.1. ATAC-Seq and RNA-Seq of Ciona Tailbud Embryos

We collected embryos at three stages (St. 21, St. 24, and St. 25; Figure 1A) to perform ATAC-seq and RNA-seq with two biological replicates. To construct the ATAC-seq library, cell isolation and nuclei extraction were performed after embryo collection (Appendix A). The embryos were then isolated as single cells. A portion of the single-cell suspension was used for activity detection. Furthermore, 0.4% trypan blue staining revealed high cell activity, and the cell viability rate exceeded 80%. After nucleus extraction, blue dot structures stained by DAPI were identified, indicating that intact nuclei were successfully extracted. Six ATAC-seq libraries were constructed, and sequencing data were obtained using an Illumina Nova Seq sequencer (Appendix A). Clean reads were aligned with the reference genome *Ciona_savignyi_51511_TX_20220104_seqkit. fasta*.

Pearson’s correlation analysis was performed for ATAC-seq data, and the results revealed high correlation coefficients between the biological repeats of St. 21 (0.96), St. 24 (0.99), and St. 25 (0.98; Appendix A). Pearson’s correlation analysis for RNA-seq data also revealed high correlation coefficients of 0.98, 0.99, and 0.97, respectively, for the aforementioned stages (Appendix A). A large number of peaks were located near the transcriptional start site, which is a regulatory area (Appendix A). This suggests that chromatin regions are involved in transcriptional regulation. Surprisingly, we also observed strong ATAC-seq signals at transcription end sites (TESs). TES open chromatin might reflect the binding of factors engaged in transcription termination [28]. All ATAC-seq libraries yielded fragment lengths with expected distributions [21,29], including most small fragments corresponding to nucleosome-free regions (≤100 bp) and mononucleosome- and dinucleosome-associated fragments (200 and 400 bp, respectively; Appendix A), confirming the good quality of the ATAC-seq data.

### 2.2. Chromatin Accessibility at Different Developmental Stages

We mapped the genomic coordinates from the peak data to the 13 chromosomes of the *C. savignyi* genome (Appendix A), and most regions of each chromosome were covered. We counted the number of peaks in different developmental stages, and the average number was 13,650 in St. 21, 31,369 in St. 24, and 19,645 in St. 25. Statistical analysis (*t*-test) revealed that the number of accessible peaks was significantly higher in St. 24 (*p* < 0.05; Appendix A), indicating that gene expression is more active during this stage. ATAC-seq peaks were enriched in 3′-UTRs, 5′-UTRs, exons, introns, promoters, and intergenic regions (Figure 1B). The promoter region accounted for the largest proportion, confirming that chromatin opening is related to gene transcriptional regulation.

The open regions of these three stages were annotated, and multiple open chromatin regions might correspond to the same gene. A total of 351 and 138 genes adjacent to differentially accessible regions (DARs) were found to contain upregulated and downregulated regions between St. 21 and St. 24 (Figure 1C, Appendix A) and between St. 24 and St. 25, respectively (Figure 1D, Appendix A). These results indicate that the chromosomal loci of ascidians exhibit time-dependent dynamics concerning chromatin accessibility. These DARs were further compared with genomic sequences, and genes adjacent to these differentially open regions were analyzed using clusterProfiler for GO enrichment (Appendix A). The results revealed that calcium ion binding, ion binding, homophilic cell adhesion via plasma membrane adhesion molecules, and transmembrane signaling receptor activity were the enriched pathways in St. 24 compared to St. 21 (Figure 1E). Compared to St. 25, the DAR-associated genes of St. 24 were enriched in the cell adhesion and exocytosis pathways (Figure 1F). Seven genes were randomly selected from the GO terms in the comparison of St. 21 and St. 24 (delta like non-canonical notch ligand 2 [*DLK2*], bone morphogenetic protein 5 [*BMP5*], bone morphogenetic protein 2 [*BMP2*], jagged 1 [*JAG*], protocadherin 18 [*PCDH18*]) and the comparison of St. 24 and St. 25 (fibrillin 1 [*FBN1*], solute carrier family 37 member 1 [*SLC37A1*]) to examine the tissue expression patterns. Although some of the GO terms were not significantly enriched, these genes belonged to genes adjacent to open regions of chromatin with significant differences. The promoter constructs of these genes were electroplated into the embryos and most of the signaling was observed in epidermal and/or muscle cells (Appendix A). These results indicate the existence of substantial dynamic regulation and function in muscle and epidermal cells during larval development.

### 2.3. Transcriptome Profiling of C. savignyi at Different Developmental Stages

Differentially expressed genes (DEGs) were obtained from RNA-seq for the three stages, and the results identified 1142 upregulated genes and 547 downregulated genes in St. 24 compared to St. 21 (Figure 2A, Appendix A). In total, 763 upregulated and 465 downregulated genes were identified in St. 25 compared to St. 24 (Figure 2C, Appendix A). Furthermore, all DEGs were mapped to GO database terms, enabling their annotation (Appendix A). DEGs were mostly enriched in the membrane transport and ion transport-related pathways (Figure 2B,D). These results suggest that in addition to basic gene expression, transcription, and translation, a wide range of processes such as ion transport, membrane transport, and cell attachment are involved in ascidian tailbud development.

### 2.4. Correlation of ATAC-Seq and RNA-Seq Data

Combined analysis of ATAC-seq and RNA-seq data can provide more biological information that can be used to investigate the relationship between changes in open chromatin regions identified through ATAC-seq analysis and corresponding changes in gene expression. In total, 70 DEGs were identified in St. 24 compared to St. 21, and 11 DEGs were identified in St. 25 compared to St. 24 (Figure 3A,B, Appendix A). In the region characterized by low chromatin openness in St. 24, 30 genes were downregulated, whereas 40 genes were upregulated (Appendix A). Meanwhile, the region with high chromatin openness in St. 25 featured two downregulated and nine upregulated genes (Appendix A). Eight genes were randomly selected from the GO terms for the comparison between St. 21 and St. 24 (Appendix A), and all eleven genes for St. 24 versus St. 25 were analyzed by qRT-PCR. Our RNA-seq and qRT-PCR analyses revealed comparable gene expression patterns (Figure 3C, Appendix A), which validated the accuracy of our RNA-seq data in all three stages.

### 2.5. Target Genes in DARs during Chordate Larval Development

To determine the tissue expression patterns of DAR-associated genes ([fh2 domain containing 1 [*FHDC1*], forkhead box f1 [*FOXF1*], connective tissue growth factor [*CTGF*], glyceraldehyde-3-phosphate dehydrogenase [*GAPDH*], rna binding motif single-strand interacting protein 1 [*RBMS1*], CTD small phosphatase 1 [*CTDSP1*], myc proto-oncogene [*MYC*], musashi rna binding protein 2 [*MSI2*], actin alpha 2 [*ACTA2*], *MMP24* and *KLF5*), we created a gene promoter construct and expressed it in *Ciona* embryos (Figure 4A). *FHDC1*, *FOXF1*, and *CTGF* were expressed in epidermal cells, and *GAPDH* was expressed in muscle cells. The expression of these genes might be associated with muscle and epidermal cell extension during early embryonic development. *RBMS1* was expressed in epidermal cells and cells near the eye spot, *CTDSP1* and *MYC* were expressed in muscle cells, *FBN1* was expressed in epidermal cells, and *MSI2* and *ACTA2* in mesenchymal cells. *MMP24* and *KLF5* were expressed in the notochord during the lumen formation stage.

Lumen formation is an obvious cytological process during the development of the ascidian tailbud. To investigate the role of *Cs*-*MMP24* in lumen formation, we used ilomastat (GM6001), a potent and broad-spectrum MMP inhibitor [30], to inhibit *Cs*-*MMP24* in *Ciona* tailbud embryos. At 12 h post fertilization (hpf), embryos were treated with 40 µM GM6001 or DMSO and then collected for phalloidin staining to mark the cell boundary (Figure 4B). We measured the A-P and D-V diameters of the extracellular lumen and calculated lumen volume (Appendix A). The results illustrated that the volume of the notochord lumen in the drug-treated group was significantly smaller than that in the control groups in St. 25 (Figure 4C). We constructed an N-terminal (activating domain) deletion construct of the *Cs*-KLF5 fusion protein (*Cs*-KLF5 DN), which retained the nuclear localization signal and the zinc finger domain, to determine the roles of *Cs*-*KLF5* in notochord lumen formation [31,32]. *Cs*-KLF5 DN expression in the notochord resulted in abnormalities in lumen formation (Figure 4D). Taken together, these results indicate that *Cs*-*KLF5* and *Cs*-*MMP24* are necessary for notochord lumen formation and expansion.

### 2.6. Motif Enrichment Analysis of DARs in the Tailbud Stage

Transcription factors (TFs) can bind accessible regions to regulate gene expression. We analyzed the enriched motifs in the different peaks in different developmental stages. To identify potential TFs responsible for the observed differences in chromatin accessibility, the enrichment of TF motifs was screened in accessible regions in St. 21 versus St. 24 and in St. 24 versus St. 25. The enrichment of the motifs is listed in Figure 5A,B. We identified 690 TFs belonging to 143 TF families (Appendix A). To investigate the mRNA expression of TFs in tailbud stages, we determined the FPKM value of each TF in St. 21, St. 24, and St. 25 by RNA-seq. The TF genes were grouped into six clusters (Appendix A). The results illustrated that *Tbx15/18/22* was upregulated from St. 21 to St. 24, and *FoxD-b*, *Six3/6*, *FoxE*, *FoxI-a*, *FoxK*, *GATA-a*, *Fos-B*, and *FoxL2* were upregulated from St. 24 to St. 25 (Figure 5A,B). *Tbx15/18/22* has been reported to maintain the expression of muscle structural genes. In *Tbx15/18/22* morphant embryos, many muscle-structural genes, such as muscle actin, myosin heavy chain, troponin t, and troponin c, are downregulated [33]. Previous studies reported that *NvSix3/6* plays a key regulatory role in the development of a broad aboral territory in *Nematostella* [34], and deregulation of the *Six3/6* pathway led to evolution of the pinhole eye in nautili [35]. The forkhead TF family is involved in various developmental processes of the embryo. *FoxD* activates vegetal hemisphere-specific genes (*Lhx3/4* and *Zic-r.b*) and represses animal hemisphere-specific genes in ascidians (*Dmrt.a* and *Dlx.b*) [36]. *FoxD* is also essential for notochord differentiation or induction [37]. *Ci*-*FoxE* is specifically expressed at the endostyle of adults, being prominent in the thyroid-equivalent region of zone 7 [38]. *FoxI* is essential for formation of the otic placode, survival of the branchial arch neural crest, and developmental remodeling of the branchial arch ectoderm [39]. *FoxK* activity is necessary for the formation of midgut constrictions and the development of midgut vesicles [40]. *FoxL2* is an evolutionarily conserved gene involved in the germinal differentiation of gonadal sex [41,42]. *GATA2* enhances the expression of *MMP2* in microvascular endothelial cells cultured within a 3D type I collagen matrix [43]. We speculated that *MMP2* is involved in notochord formation. *Fos-B* overexpression resulted in aberrant axon branching in zebrafish [44].

In conclusion, we identify 15 TFs responsible for the observed differences in chromatin accessibility. The expression profiles of these TFs provided a possible clue for their functions (Figure 5C).

## 3. Discussion

Gene expression is regulated by changes in gene regulatory programs through the integration of Cis-regulatory elements (CREs), which include core promoters, promoter proximal elements [45], and TF binding sites. The binding of TFs to CREs promotes the recruitment of RNA polymerase II to the promoter of the target gene, thereby regulating the gene expression [46]. Identification of the sequences of CREs and clarification of their localization on chromosomes and binding TFs will be helpful in understanding the epigenetic information of chromatin and its molecular mechanisms in a specific biological context [46,47]. In addition, transposable elements (TEs) may also have a significant role in organizing chromatin structure and gene regulation, which contribute to controlling the expression of genes locally and even far away at the transcriptional and post-transcriptional levels [48]. In a study of *Zootoca vivipara*, TE was found to be highly conserved and played a positive role in scaly animal genome evolution [49]. In this study, we screened key regulatory genes in the complex regulatory network through high-throughput sequencing and revealed the molecular regulatory mechanisms of the development of ascidian larvae.

Our ATAC-seq results identified vital genes associated with notochord tubulogenesis and lumen expansion, including *DLK2*, *BMP5*, *BMP2*, *JAG*, *PCDH18*, *FBN1*, and *SLC37A1*. *DLK2* belongs to the Notch/Delta/Serrata family, and it has been reported to regulate metastatic melanoma cell proliferation in a dose-dependent manner [50]. JAG1 is the ligand for Notch1, and upregulated *JAG1* promotes cell proliferation in adrenocortical carcinoma [51]. *BMP2* [52] and *BMP5* [53] have been reported to regulate cell proliferation. They target epidermal and muscle cells, and they might be involved in regulating the proliferation of epidermal and muscle cells used for tail extension. It has been reported that BMP signaling regulates anterior neurectoderm specification and differentiation in *C. intestinalis* [54]. *PCDH18* expression was observed near the eyespot and CNS, which is similar to its localization in zebrafish [55]. Diversified PCDH molecules contribute to neural circuit development. In in vivo and in vitro mouse hippocampal neurons, Pcdha knockout resulted in the decreased dendrite arborization, and this arborization defect was dependent on a PYK2/Rho/Rac pathway [56,57,58]. From St. 24 to St. 25, *FBN1* was expressed in epidermal cells, and *SLC37A1* was expressed in notochord cells. They might be involved in epidermal cell extension and notochordal lumen formation and expansion. The RNA-seq results identified DEGs enriched in membrane transport and ion transport-related pathways, consistent with the formation of the notochordal lumen through the establishment of polarity, cytoskeletal rearrangement, and expansion of the notochordal lumen with ion flow (SLC26A), proteoglycan secretion, vesicle trafficking (ELMOD3), and replenishment of the apical membrane (14-3-3εa and caveolin) [17,59,60,61,62].

A comprehensive analysis of ATAC-seq and RNA-seq data demonstrated that all 12 DEGs had high chromatin openness in St. 25. The analysis revealed that increased chromatin accessibility was correlated with increased transcription, although there was a clear increase in regional accessibility corresponding to decreased transcription, which might be attributable to transcriptional repressors binding to open chromatin regions [63]. Among them, *MMP24* and *KLF5* targeted notochord cells in the lumen formation stage. The dominant-negative effect of *Cs*-*KLF5* and drug treatment targeting *Cs*-*MMP24* resulted in abnormal lumen formation and expansion. *KLF5* induces ECM degradation by promoting MMP expression and secretion [64]. *KLF5* has been identified as a crucial factor in cardiovascular diseases [65]. We hypothesize that *Cs*-*KLF5* and *Cs*-*MMP24* coregulate the formation of the notochord lumen.

Motif enrichment analysis of open regions revealed enriched TFs, including *Tbx15/18/22*, *FoxD-b*, *Six3/6*, *FoxE*, *FoxI-a*, *FoxK*, *GATA-a*, *Fos-B*, and *FoxL2*. The potential roles of these TFs were summarized according to the functions of these genes in ascidian larval development (Figure 5C).

Overall, we identified 12 candidate genes through a combination analysis of ATAC-seq and RNA-seq data. *Cs*-*KLF5* and *Cs*-*MMP24* were proven necessary for the formation and expansion of the notochord lumen. Our approach for identifying open chromatin regions and predicting TFs involved in the regulation of ascidian development offers new resources for further mechanistic exploration in ascidian embryonic development.

## 4. Materials and Methods

### 4.1. Animals

Adult specimens of *C. savignyi* were collected from Huangdao, Qingdao, China. The animals were kept in the laboratory under controlled conditions, in seawater with a salinity of 32‰. We obtained eggs and sperms through dissecting adult animals. After fertilization and dechorionation, embryos were cultured at 16 °C. We collected embryos at three different stages for ATAC-seq and RNA-sequencing.

### 4.2. ATAC-Seq Analysis

Raw sequencing reads were initially processed and adapter sequences were removed using Fastp [66]. Clean reads were obtained from the raw data by quality control and then aligned to the reference genome *Ciona_savignyi_51511_TX_20220104_seqkit. fasta* (unpublished data) with Bowtie2 [67] using default settings. MACS2 software (v2.2.7.1) [68] was used to detect peaks. The parameters of peak calling are --nomodel, --shift -100, --extsize 200. According to the genomic location and gene annotation information of peaks, peak-related genes can be annotated using ChIPseeker [69]. Moreover, the distribution of peaks in different genomic regions (such as exon, intron, promoter, downstream regions, intergenic regions, 3’-UTRs, 5’-UTRs) was estimated. Promoters are defined as 2 kb upstream and downstream of TSS. Differentially accessible chromatin regions (DARs) were identified by DiffBind software (v3.8.4) [70]. Peaks with FDR under 0.05 were considered differentially accessible peaks. And the R-package ChIPseeker was used to associate differentially accessible peaks with their putative target genes [69]. Transcription factor enrichment of sequences from the known urochordate motif database from JASPAR was performed using SEA (Simple Enrichment Analysis, evalue ≤ 10) [71]. Downstream gene prediction was performed by motif scanning of open chromatin regions using FIMO.

### 4.3. RNA-Seq Analysis

RNA-seq reads that passed the filter were trimmed to eliminate low-quality reads and adapters by Fastp [66]. Reads were aligned by STAR [72]. The edgeR method has high sensitivity and accuracy for medium-scale RNA sequencing data, so differential gene expression was analyzed using edgeR [73]. Significantly differentially expressed genes (DEGs) were identified based on the following criteria: |log2 fold change| ≥ 1 and padj < 0.05. Two biological replicates were used. The R package ggplot2 was used to plot heatmap and volcano plots [74].

### 4.4. GO Enrichment Analysis

GO enrichment analysis of DEGs and genes adjacent to DARs was implemented using the clusterprofiler (v4.6.2) [75] R package. GO annotations came from annotations on *C. savignyi* genomes (unpublished data) using interproscan. GO terms with a corrected *p*-value < 0.05 were considered significantly enriched.

### 4.5. Integration Analysis of ATAC-Seq and RNA-Seq

We compared the expression profiles of differentially accessible peak-related genes in ATAC-seq with DEGs in RNA-sequencing. Genes associated with closed-to-open peaks in ATAC-seq were compared with DEGs upregulated in RNA-seq. Genes associated with open-to-closed peaks in ATAC-seq were compared with DEGs downregulated in RNA-seq [76]. Genes with changes in both gene expression and chromatin accessibility were shortlisted, followed by Spearman’s correlation analysis.

### 4.6. Identification and Classification of TFs

The methods used here follow a previously described protocol [77]. TFs were identified and categorized according to the conserved DNA binding domains (DBDs). The protein domains were analyzed using Hidden Markov Model (HMM) profiles from the Pfam database [78], applying the hmmscan program in the HMMER package. Pfam ID of DBDs were identified from the REGULATOR database [79]. Genes containing DBDs were screened and classified as TFs with an e-value threshold of 10^−4^. Pfam IDs are shown in Appendix A. TFs were classified into different families according to the DBDs.

### 4.7. Quantitative Real-Time PCR (qPCR)

These methods are described in detail in [80]. RNAiso plus (TAKARA, Kusatsu, Shiga Japan) was used to isolate total RNA from frozen embryos at different stages. For reverse transcription, HiScript II Q RT SuperMix for qPCR (+gDNA wiper) was utilized. The expression levels of DEG/DAR-associated genes were detected by 2-step qPCR (Vazyme, Nanjing, China). We calculated data using 2^−ΔΔCt^. Statistical analysis was conducted using paired Student’s *t*-tests, with a significance level of *p* < 0.05. Prism 9 software (v 8.0.2) was used to plot the graph of qPCR results. The primers used in qPCR are listed in Appendix A.

### 4.8. Plasmid Constructions

Tissue expression patterns were performed by gene promoter analysis. DNA sequences upstream of DEG/DAR-associated genes were amplified and subcloned into the pEGFP-1 vector for promoter analysis. And the promoter sequence was compared to the genes encoded in the genome through BLAST [81] to ensure that the promoter did not contain other genes. Then, these constructs were confirmed by sequencing. The primers used in plasmid construction are listed in Appendix A.

### 4.9. Electroporation

We performed electroporation experiments according to a previously described method with some modifications. Specifically, the parameter of the exponential protocol was used with 50 V and 1500 μF [82]. The electroporation system consisted of a plasmid of 60–80 μg, 420 μL 0.77 M D-mannitol, and ddH_2_O. Dechorionated fertilized eggs (300 μL) were mixed with the electroliquid system in 0.4 cm cuvettes of a Gene PulserX cell System (BIO-RAD, Hercules, CA, USA).

### 4.10. Drug Treatment and Immunostaining

Embryos were treated with 40 µM of the inhibitors ilomastat (GM6001, HY-15768, MedChemExpres, Monmouth Junction, NJ, USA) [83] and DMSO at 12 h post fertilization (hpf). Up to 22 hpf, embryos were collected and fixed with 4% paraformaldehyde (PFA) at room temperature for 2 h. Fixed embryos were washed three times with PBS containing 0.1% Triton X-100 (PBST), and then stained with phalloidin in PBST (1:300) overnight at 4 °C to mark the cell boundary of embryos. Embryos were then washed three times with PBST. DAPI staining was performed in the dark to label the cell nucleus.

### 4.11. Statistics

Lumen volume was measured according to the method described previously [84]. We also measured the anteroposterior (A-P) diameter (d1) and dorsoventral (D-V) diameter (d2) using ImageJ and calculated lumen volume using the spheroid formula (V = ⅙πd1d2^2^) (Appendix A). Then, we counted the lumen volume in the DMSO and inhibitor treatment group separately. All data are presented as mean ± SEM. We performed statistical analysis using Student’s *t*-test. *p* < 0.05 (*) and *p* < 0.01 (**) indicate statistical significance.

## Figures and Tables

**Figure 1 ijms-25-02793-f001:**
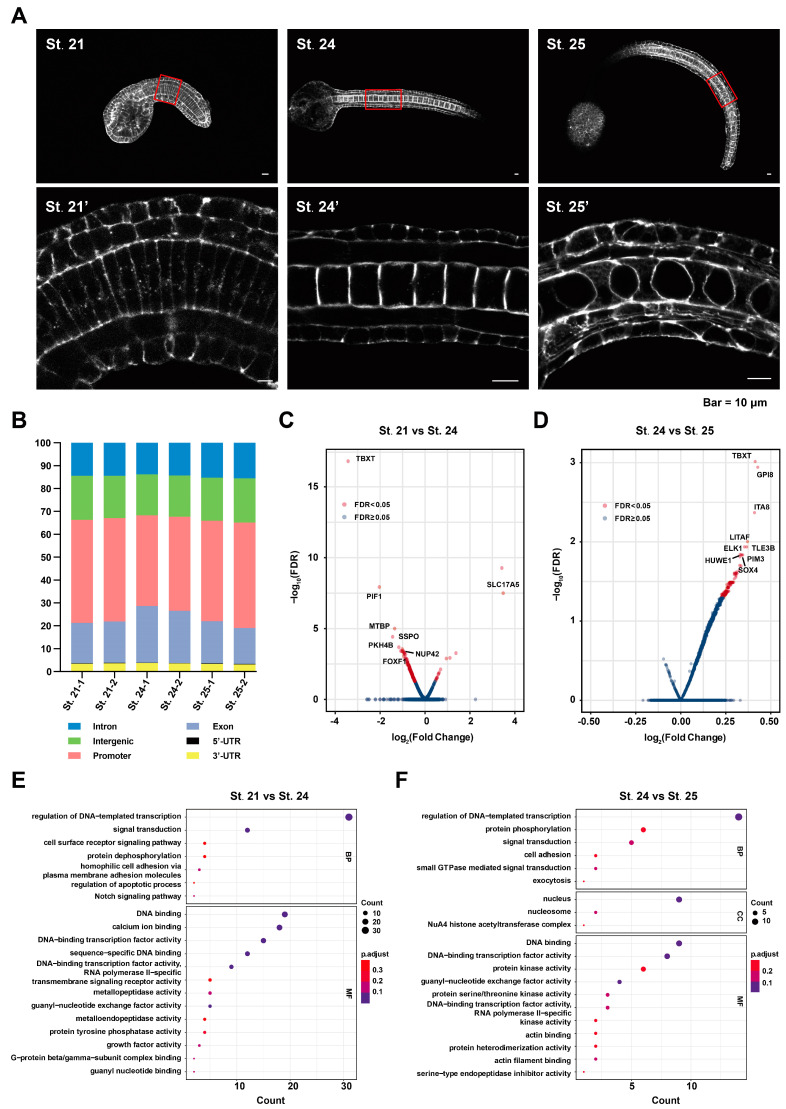
Morphological characterization of embryonic development in *C. savignyi* and the landscape of DNA accessibility in three developmental stages. (**A**) St. 21, St. 24, and St. 25 *C. savignyi* embryos cultured at 16 °C were stained with phalloidin. The upper images present the intact ascidian embryo, and the lower images present the notochord tissue. Scale bar, 10 μm. (**B**) Peak distribution ratio of gene functional elements. Six regions were detected: 3’-UTRs, 5’-UTRs, exon, intron, promoter, and intergenic regions. (**C**,**D**) Scatter plot indicates the pattern of differentially accessible regions (DARs) in different developmental stages, including both FDR < 0.05 (red) and FDR ≥ 0.05 (blue) genes. (**E**,**F**) GO enrichment analysis of DAR-associated genes. C and E: St. 21 vs. St. 24; D and F: St. 24 vs. St. 25.

**Figure 2 ijms-25-02793-f002:**
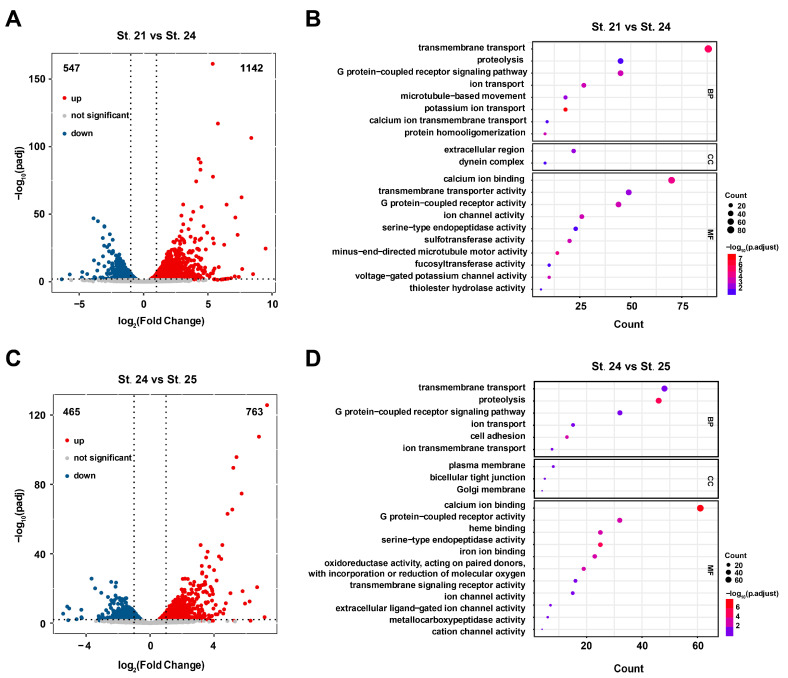
Visualization of DEGs. (**A**,**C**) Visualization of gene expression changes. padj is represented by −log_10_(padj) in the ordinate, and gene expression is represented by log_2_ (fold change) in the abscissa. Blue, red, and gray dots indicate downregulated DEGs, upregulated DEGs, and genes with no significant expression change, respectively. (**B**,**D**) GO enrichment analysis of DEGs at different stages. B: St. 21 vs. St. 24, D: St. 24 vs. St. 25.

**Figure 3 ijms-25-02793-f003:**
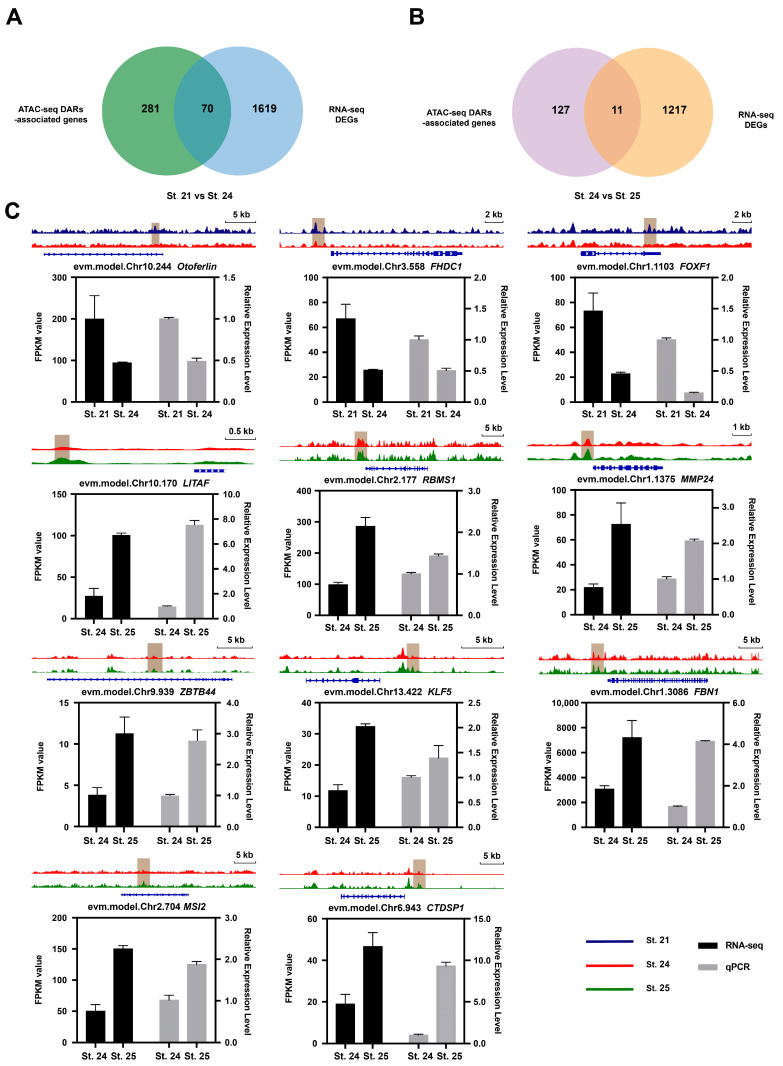
Integrated analysis of ATAC-seq and RNA-seq data. (**A**,**B**) Intersection of different mRNAs and target genes of different peaks. DAR-associated genes (green and purple indicate St. 21 vs. St. 24 and St. 24 vs. St. 25, respectively) and DEGs (blue and orange indicate St. 21 vs. St. 24 and St. 24 vs. St. 25, respectively). (**C**) Chromatin opening and expression changes of associated genes. IGV plots present ATAC-seq signals in selected gene loci. Gene expression was determined by quantitative fluorescence PCR and transcriptomic analysis. Blue, red, and green represent chromatin fragments of St. 21, St. 24, and St. 25, respectively. Black and gray bars represent transcriptome data and quantitative fluorescence PCR results of genes, respectively.

**Figure 4 ijms-25-02793-f004:**
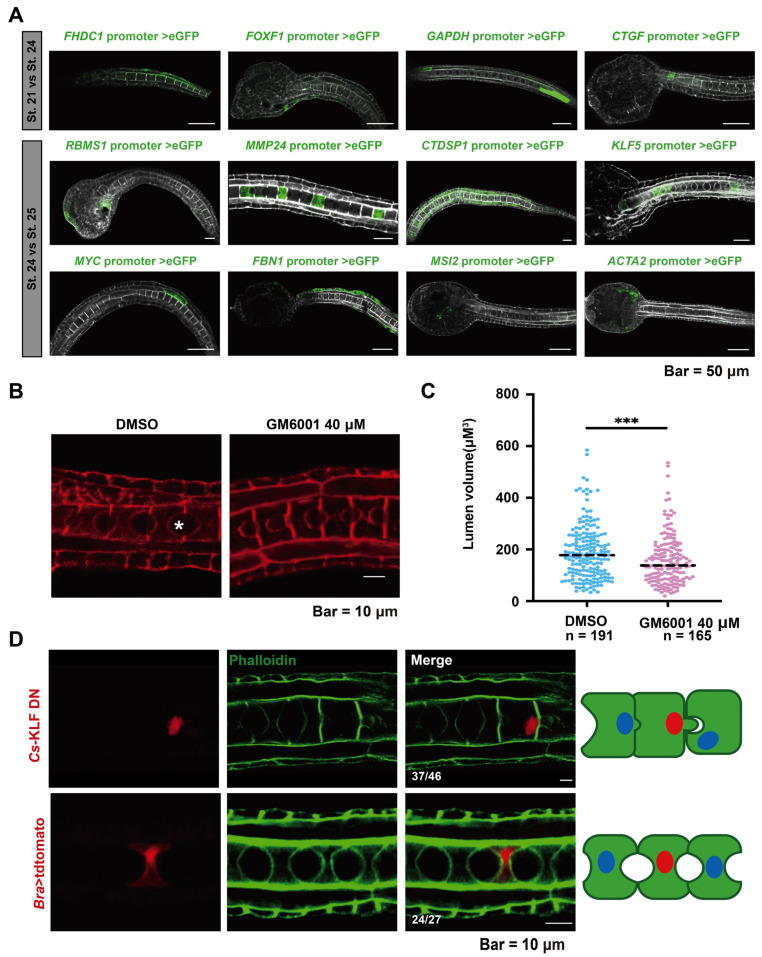
Expression and functional assay of key genes in larval development. (**A**) Tissue expression patterns of the selected genes as determined by promoter assay. Scale bar, 50 μm. (**B**) Treatment with an *MMP24* inhibitor (GM6001, 40 μM) arrested lumen expansion. Scale bar, 10 μm. The asterisk marks the lumen. (**C**) Quantification of lumen volume in DMSO- (n = 191) and GM6001-treated embryos (n = 165) embryos at 22 hpf. Statistical analysis was performed using Student’s *t*-test, *p* < 0.001 (***). (**D**) Overexpression of dominant-negative *Cs*-KLF5 led to failure of lumen formation. Schematic images located on the right side of confocal images present the phenotypes of *Cs*-KLF5 knockdown and control. The green, blue, and red areas represent the notochord cells, nuclei, and nuclei in cells expressing *Bra*>*Cs*-KLF5DN::tdtomato/*Bra*> tdtomato (red), respectively. Scale bar, 10 μm.

**Figure 5 ijms-25-02793-f005:**
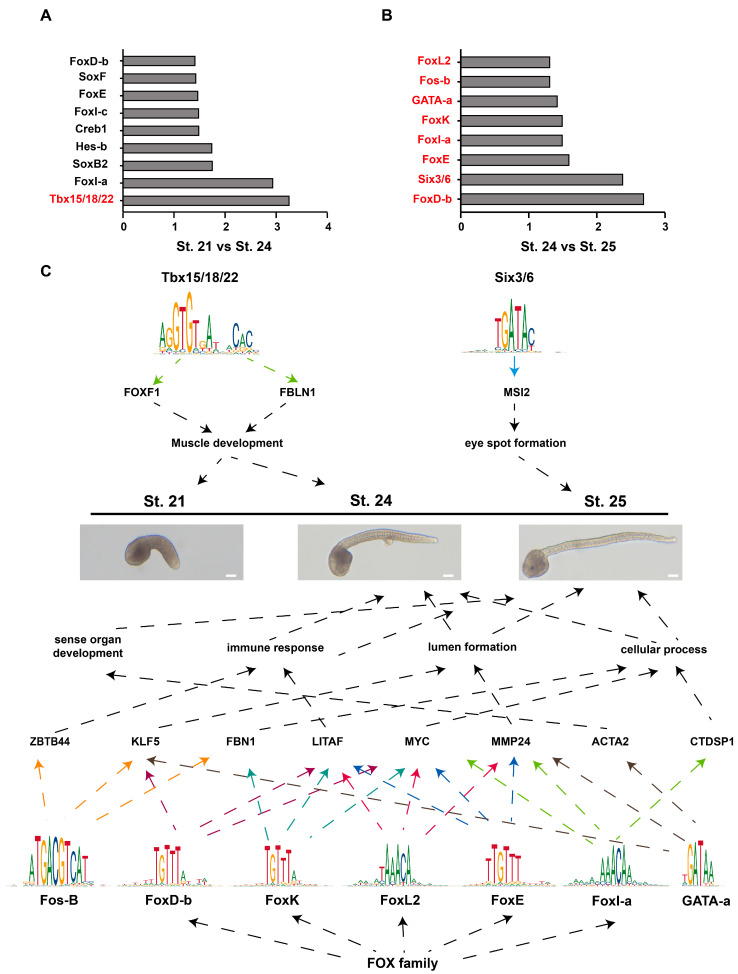
TF binding motif enrichment analysis. (**A**) Nine TF binding motifs enriched in accessible regions from closed to open in St. 21 versus St. 24. (**B**) Eight TF binding motifs enriched in accessible regions from closed to open in St. 24 versus St. 25. Red highlights the upregulation of TFs. (**A**,**B**). The *X*-axis represents −log10 (*p*-value) in the motif enrichment analysis. The *Y*-axis represents the predicted TF binding motifs. (**C**) Summary of the potential roles of TFs in *Ciona* larval development. Different colored arrows represent downstream genes that may be regulated by different transcription factors.

## Data Availability

The datasets presented in this study can be found in online reposito- ries. The ATAC-seq data of all stages and RNA-seq data of St. 21 and St. 24 are deposited in NCBI under the BioProject number PRJNA1064177. The RNA-seq data of St. 25 were downloaded from the NCBI SRA database, with access numbers SRR6431994 and SRR6431995.

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
