# Peer review of "Dynamics of Chromatin Opening across Larval Development in the Urochordate Ascidian *Ciona savignyi"

_ijms, 2024, doi:10.3390/ijms25052793_

Round 1
Reviewer 1 Report
Comments and Suggestions for Authors
In this study He et al studied the mechanisms of the development of ascidian embryos combining ATAC seq and RNA-seq. Methods and thresholds used for data analysis are adequate, however not consistently applied.
The topic of this review is of potential interest. Nevertheless, the following points have to be addressed by the authors.
Major concerns
1. In the abstract the authors stated a ‘strong correlation between gene expression and chromatin accessibility’. This was based on which analyses? Please explain these in the methods part. If it is based on figure3 A/B this doesn’t seem a high correlation to me.
2. In suppl. Tab3 and 4 are 2 sub-tables: ‘*_deseq2.peak.anno’ and ‘Differentially peak’. Is the differentially expression calculated by DESeq2 (default method in DiffBind)? Why did the authors use edgeR for differential expression in mRNA seq? For better comparability it would be cleaner to use the same method.
3. Table S3: All peaks and DARs of ATAC-seq in St. 24 compared to St. 21: As far as I understand, in columns G and H log2 read counts for the respective peak are given. What is Conc (column F). It does not seem to be DESeq2 baseMean.
4. L105ff: ‘The open regions of these three stages were annotated, and 351 and 138 differentially accessible regions (DARs) were found to contain upregulated and downregulated regions between St. 21 and St. 24 4 (Figure 1C, Table S3) and between St. 24 and St. 25, respectively (Figure 1D, Table S4)’. Counting all peaks from ‘St 21-St 24--D_deseq2.peak.anno’ with FDR<0.05 I get 447 DARs, in S4 I get 157. Do you mean ‘genes’ instead of ‘DARs’?
5. L332: .GO terms with a corrected P-value < 0.05 were considered significantly enriched’. In figure1 E/F GO terms with a corrected p.value <0.3 are shown. Further, the GO-terms mentioned in lines 111-115 are partially not significantly enriched according to the definition in the methods part (e.g. ‘transmembrane signaling receptor activity’: p.adjust= 0.3048557). Please clarify. The selection of the mentioned GO-terms seems rather random.
6. Figure4: ‘. Data are presented as the mean ± SEM’. This is a stripplot. I don’t see any SEM. Please correct.
7. L336ff: ‘The genes associated with the upregulated peaks in ATAC-seq were compared with the DEGs upregulated in RNA-seq. The genes associated with the upregulated peaks in ATAC-seq were compared with the DEGs upregulated in RNA-seq’. If you use FDR for ATAC data only, how is up-/downregulation defined? Equal to mRNA seq with an additional threshold for logFC (in this case I get 4 up- and 15 downregulated in S3 and none in S4, which would be in accordance with Figure1 C/D)? Please clarify in the methods part.
8. Methods, Identification and Classification of TFs: I do not understand this section. Please rewrite this to make it clearer.
9. L289: ‘FIMO was used to predict downstream genes’ Please include in the methods part.
10. L82ff: Please explain, why for ATAC-Seq you used Spearman’s rank test and for RNA-Seq Peason Correlation.
11. Figure5: . Red highlights the upregulation of TFs. (A and B)’. I don’t see red.
Reviewer 2 Report
Comments and Suggestions for Authors
This manuscript deals with pathways of chromatin opening, gene expression and regulation in the development of Ciona savignyi.
This work deals with poorly studied mechanisms in the development of the studied species. It is overall well written, the methods are adequate and the results should deserve publication.
However, there are also some issues that lower the quality of the work and the authors should carefully deal with them before the manuscript can be accepted for publication.
In particular, the presentation of images should be reorganized, some methods should be better described and the Introduction and the Discussion need to be broadened including relevant citations on what is already known on some regulatory mechanisms in Ciona, the genome and the chromosome complement of the study species and the possible role of transposable elements in organizing chromatin structure and gene regulation (see below.
Figures are of high quality, but Fig F1A and Fig 1B-F should be separated. In fact, the graphs should be easier to read. It would probably be better to split Fig. 1 into 3 distinct figures.
The same is true for Figures 2 and 3. Please, make those images more reader friendly .
Please check the manuscript for typos and some sentences that are not very clear.
In the Title, the species name should be in lowercase.
Introduction
Given the research topic I suggest to describe more in detail in the introduction the overall genome structure and chromosome complement of the study species, adding relevant citations.
It would also be very useful to specify in the Introduction or the Discussion that transposable elements (TEs) may have a significant role in organizing chromatin structure and gene regulation (Petraccioli et al. 2019 Cytogenetic and Genome Research 157, 65-76, Gebrie 2023 Mobile DNA 14, 9).
Line 35: Urochordata = Tunicata should be considered a subphylum. Please, specify with relevant citations the taxonomy/phylogeny you are referring to.
Lines 56-60: Please, explain why these stages were selected.
Lines 43-45: The authors can describe here what is already known on molecular mechanisms
Results
Line 227: Which genes?
Line 240-241: I suggest removing this sentence or to move it to the Discussion.
Discussion
line 270: Plese, be more specific.
Methods
Line 301. How many adults were collected and how many samples were used in each protocol?
Line 301: How the authors performed a taxonomic identification of the study samples?
Line 324: In general, I think it would be better to adjust some parameters. But the authors could also explain why they used default parameters.
Line 325: correct “by performed using”
Line 363: Please, describe the modifications
Comments on the Quality of English Language
English is fine, but there are some typos and unclear sentences.
Round 2
Reviewer 1 Report
Comments and Suggestions for Authors
The manuscript has been significantly improved. I would recommend it for publication.
Reviewer 2 Report
Comments and Suggestions for Authors
The authors have done a good job in revising their work, adjusting the images, which are now more readable, and improving several section of the manuscript.
The manuscript can now be accepted for publication in my opinion.
Comments on the Quality of English Language
There are still some typos and sentences that should be corrected, but overall also the english has been improved and can be further checked during proof stage.
e.g. Line 353 These methods as the previously described protocol [77].